# Synthesis, Characterization and Development of Energy Harvesting Techniques Incorporated with Antennas: A Review Study

**DOI:** 10.3390/s20102772

**Published:** 2020-05-13

**Authors:** Husam Hamid Ibrahim, Mandeep S. J. Singh, Samir Salem Al-Bawri, Mohammad Tariqul Islam

**Affiliations:** 1Department of Electrical, Electronic and Systems Engineering, Faculty of Engineering and Built Environment, Universiti Kebangsaan Malaysia, UKM, Bangi 43600, Malaysia; hussampc93@gmail.com (H.H.I.); mandeep@ukm.edu.my (M.S.J.S.); 2Department of Electronics & Communication Engineering, Faculty of Engineering & Petroleum, Hadhramout University, Al-Mukalla 50512, Hadhramout, Yemen; s.albawri@gmail.com

**Keywords:** energy harvesting, radio frequency, rectenna, antenna, rectifier monopole antenna

## Abstract

The investigation into new sources of energy with the highest efficiency which are derived from existing energy sources is a significant research area and is attracting a great deal of interest. Radio frequency (RF) energy harvesting is a promising alternative for obtaining energy for wireless devices directly from RF energy sources in the environment. An overview of the energy harvesting concept will be discussed in detail in this paper. Energy harvesting is a very promising method for the development of self-powered electronics. Many applications, such as the Internet of Things (IoT), smart environments, the military or agricultural monitoring depend on the use of sensor networks which require a large variety of small and scattered devices. The low-power operation of such distributed devices requires wireless energy to be obtained from their surroundings in order to achieve safe, self-sufficient and maintenance-free systems. The energy harvesting circuit is known to be an interface between piezoelectric and electro-strictive loads. A modern view of circuitry for energy harvesting is based on power conditioning principles that also involve AC-to-DC conversion and voltage regulation. Throughout the field of energy conversion, energy harvesting circuits often impose electric boundaries for devices, which are important for maximizing the energy that is harvested. The power conversion efficiency (PCE) is described as the ratio between the rectifier’s output DC power and the antenna-based RF-input power (before its passage through the corresponding network).

## 1. Introduction

There are no limitations to the wireless transmission of energy. Wireless energy transfer requires wireless power transmission through a transmitter linked to a power supply; the energy is converted back to an electric current, and later it is used to transfer the energy back to one or more receivers. The use of affordable, green communication strategies has become crucial due to the emergence and increasing popularity of the Internet of Things (IoT) and implementations of broad-scale wireless networks with sensors. Most of these devices need to function without batteries, and for this reason, an energy harvesting circuit is required so that they can capture wireless power [1,2].

Three subsystems can be grouped into an energy harvesting system. The first subsystem collects energy by using coils that allow the inductive communication or antennas, which enables the far-reaching transmission of power. Once the energy is collected, its harvested type will not be usable, and a second subsystem is thus needed to convert the energy. The second subsystem alters the captured energy to make it usable. Here, the use of a rectifier is employed to convert the received AC signal to DC, because the majority of devices need a DC power supply. The proper storage of the rectified energy must be ensured so that it can be used in the future. The third subsystem is responsible for storing the energy obtained through the wireless energy harvester, and this subsystem is of great importance because of the availability of low-input power [1,3,4].

The efficiency of the harvested energy can be improved and maximized through the optimization of the three subsystems. With regards to the energy propagation to the harvester, power can be efficiently transferred when an appropriate scheme is selected. The correct scheme relies on the range from the transmitter (Tx) to the receiver (Rx). Magnetic-induced coupling transfers power to the reactive close-field area. When there is a greater distance, the transfer of power into the far-field is performed using antennas [1,5]. The use of an antenna array can be employed to improve the efficiency of the system in each of the operating regions; this is because the antenna array has the capacity to scan and focus beams. In order to further improve the energy storage and the rectifier, which are the other subsystems, an appropriate rectifying element and topology must be used alongside a suitable energy storage device for the application [3]. The main reason for the use of energy harvesting technologies is to enable the creation of a wireless sensor platform that is totally independent and self-sustainable. These technologies can also be used for the creation of power-efficient devices for smart cities, smart homes, and IoT applications that could significantly influence our future lives. Nevertheless, the optimal design of a rectifying-antenna—or rectenna—remains a challenge [6].

The intelligence of factories, offices, and homes is increasing due to the emergence of IoT devices as well as wireless sensors, all of which need to be powered by electricity. Even though a battery is often employed in these devices, there is still a need for electricity because the batteries require maintenance and have a limited life span [6]. In an attempt to extend the life span of the batteries, wireless charging solutions have been developed by researchers; some of the charging solutions include RF power harvesting, solar power harvesting, and inductive charging. In RF power harvesting, RF energy is captured from ambient RF sources such as Wi-Fi hotspots and cellular base stations, and the collected RF energy is then converted into DC power. The rectification device, consisting of two major components (an antenna and a conversion circuit (RF-to-DC)), accomplishes this procedure [6]. The sources of ambient RF energy include FM, Wi-Fi, Global System for Mobile (GSM), Digital TV, AM, Bluetooth, WLAN, etc. There are different devices that are powered by ambient RF energy, including wireless body area networks, radio frequency identification tags, wireless sensor nodes, etc. [7].

An antenna is located at the front end of an RF energy harvester, and the RF signals are captured and delivered to the other circuit elements by this antenna. The amount of harvested energy depends on the radiation parameters of the antenna, such as the beam width, efficiency, gain, polarization, bandwidth, etc. Basically, an RF energy harvesting (RF-EH) antenna must be portable, compact, integrable, etc. to enable easy embedding. Furthermore, the antenna must possess high gain in order to enable the reception of available energy, and it must also possess high efficiency so that losses can be minimized [8]. Several studies have reported the harvesting of RF power from a single frequency band [8,9,10]. If the usable power range is low, a minimum output capacity is provided. In fact, it is essential to obtain energy from a wide-frequency band or multiple bands, which allows more DC power to be harvested. Nevertheless, because of the nonlinearity of the rectifier, the architecture of multiband or wideband rectennas is rather challenging. In [11], a U-slot on a patch was used to graft a broadband antenna. A T-shaped feeding patch and an inverted T-shape stub were used to produce the four-band antenna in [12]. A wide-band antenna and double-band were developed in [13]. In [14], the authors present a wideband antenna produced by a dielectric Minkowski fractal resonator. Furthermore, the antenna could be operated in multiband by inserting a T-shaped slot in the ground plane [15]. For broadband RF energy harvesting, an improved antenna bandwidth is not adequate, as the rectifier will run on the same frequency band as the antenna. Multiband and wireless rectennas have been introduced with different techniques. In [16,17], the authors documented the use of several correction devices working with a single or dual-band.

This article provided a review of energy harvesting development procedures from recent publications. Moreover, a comparison between energy harvesting techniques is provided based on different energy sources and parameter characteristics. Nevertheless, several kinds of antennas that might be used in comprehensive energy harvesting systems are reviewed prior to a comparison in terms of their size, gain, efficiency, etc. Furthermore, rectennas systems have been studied as reported in the state-of-the-art systems.

## 2. Energy Harvesting

Energy harvesting is a process by which small quantities of obtainable energy are harvested or collected from the atmosphere and turned into power, which can be utilized instantly and later. In this process, the use of an energy source is extended to locations in which grid power is absent [18]. The harvested energy is sufficient for most wireless applications, remote sensing, Radio Frequency Identification (RFID), and body implants. In addition, the life span of batteries can be extended using most sources of energy harvesting, even if the gathered energy is low. Most low-power devices, such as medical devices, sensors, and portable devices, are largely powered by batteries, which must be replaced every 5–10 years due to their limited lifespan [19]. The main purpose of harvesting energy is to enable the conversion of energy from one form to another so that it can be utilized to power electronic devices. When the implementation of energy harvesting is carried out in environmental monitoring nodes, ambient energy can be directly extracted from the environment under surveillance, and then the extracted power can be used to power environmental wireless sensor network (EWSN) nodes. Thus, the performance of the EWSN nodes can be improved while extending their lifetime. Due to the opportunities offered by outdoor environments, natural elements such as sunlight or wind present in the environment can be exploited. In addition to these natural elements, nodes can be powered using other kinds of energy such as RF signals emanating from human activities [20].

A wide range of energy sources has been considered for energy harvesting. One of the main criteria that must be considered when selecting a source of energy harvesting is its capability to provide the needed power level for the sensor node. Generally, the dissipation of power occurs when voltage is being converted, and the increase in the voltage input–output ratio causes the power dissipation to increase. Thus, it is crucial to ensure that the voltage and current of the generated power are appropriate. The desired power level can be achieved by increasing the energy source or by scaling the energy harvesting device accordingly [21]. In the work of Wang et al. [22], the researchers introduced a lightweight double-band rectenna for power and energy storage for wireless transmission. They integrated a new antenna which resembled a vine, featuring a dual-band network combining two separate radial branches and a full Greinachar rectification circuit. With the rectenna, it became possible to capture and transform RF capacity concurrently with 2.45 GHz Wi-Fi and 3.5 GHz Worldwide Interoperability for Microwave Access (WiMAX) bands. In another study, a rectifier-less AC–DC energy harvesting circuit was proposed by Shaikh and Zeadally [23]. With the proposed energy harvesting circuit, energy could be harvested from different low-voltage piezoelectric transducers. Shi, Fan, Li, Yang and Wang [24] proposed a compact broadband slotted antenna in order to increase the efficiency of rectennae in harvesting RF power from the LTE-2300/2500 band. In the study carried out by da Silva, Neto, and Peixeiro [25], a fast and accurate method was proposed, and they employed only the *Z_ant_* in the desired operating frequency rather than the whole touchstone model. The aim of using only the *Z_ant_* in the desired frequency was to increase the simplicity of the proposed method. 

In another study, a multiport rectenna was proposed by Shen, Zhang, Chiu, and Murch [26]. The multiport rectenna was proposed for the harvesting of ambient RF energy. The novelty of their work lies in the fact that the number of ports used depends on the frequency when the same overall area is used. Awais et al. [27] presented a miniaturized slotted planar antenna alongside a system for energy harvesting. The proposed system has a compact size of 18 mm × 30 mm with broadband characteristics, and it offers a better gain for different frequencies. In the work carried out by Yang et al. [28], the authors introduced ta version of the self-sustained body area network (BAN) sensor. The model consisted of a compact electro-triple-band rectenna, micro controls, direct current (DC) regulation, and the sensing module, as well as the contact module and the storage module. A new cube with dual-band antenna arrays was proposed by Zhu, Zhang, Han, Xu, and Bai [29] for RF energy harvesting. The bandwidth of the proposed harvesting cube ranged from 1.85 to 1.93 GHz and from 2 to 2.10 GHz, respectively, covering GSM-1800 and Universal Mobile Telecommunications System (UMTS-2100) with cell networking bands. Zhang et al. [30] proposed and designed an RF energy harvesting system and RF-based wireless power transfer system for applications that are self-sustaining and battery-less. An overview of a reported studies of energy harvesting is summarized and demonstrated in Table 1. The following are the key aims of energy harvesting technology:Increase of lifespan;Removal of main supply wires;Elimination or reduction of dependence on batteries;Maintenance or/and performance enhancement;Ease of installation;Reduction of expense;Reduction of waste.

## 3. Micro-Scale Energy Harvesting Classification

Energy scavenging is aimed at the extension of battery life or the replacement of the battery. Nevertheless, the creation of a device which can effectively scavenge energy with an energy output that is large enough to support a sensor network is complicated. A variety of factors could be used to determine the suitability of the corresponding energy for a given wireless body area network (WBAN) application. Some of the parameters that should be put into consideration include the output power, the system’s physical size, output impedance, and the availability of the ambient source. In particular, the output power of an energy scavenging circuit ranges from micro-watts to milli-watts, and the required power to run a system is comparatively low. Besides, losses are caused by complex systems with several interface circuits, and such losses result in reduced circuit efficiency. Thus, one of the common approaches used in increasing the efficiency of these circuit interfaces is to combine them; by combining them, the size and cost of the circuit can be reduced [31].

Nevertheless, the integration of circuit interfaces poses a new challenge for a circuit designer in terms of cost and size constraints. Thus, for WBANs and other potential applications, the integrating level for an energy scavenging circuit is another significant parameter. This section presents the classification of methods used for the scavenging of micro-scale energy based on the kind of source shown in Figure 1. 

### 3.1. Photovoltaic (PV) Energy Harvesting

In photovoltaic (PV) energy scavenging, electricity is produced by devices by using sunlight or any other artificial source of light. Usually, the light from an external source is absorbed by the PV cells, which are made up of semiconductor materials. Due to the effect of the p-n junction, absorbed light releases electrons from the semiconductor. The collection of the released electrons and holes occurs at the electrodes, creating a voltage difference [31].

However, the major disadvantage of this technology is that it is not functional without a light source, and as such, the number of applications is limited [32]. The use of photovoltaic technology has been successfully employed for commercial use in both small-scale energy harvesting and large-scale renewable energy industrial applications. In small-scale industry, it is used in the form of leisure trickle chargers and portable power banks. In contrast, in large-scale industry, it is applied in auxiliary power generation and solar farms. In recent years, researchers have focused on developing organic thin-film solar cells (OTFSCs), which are comparatively cheap and require less energy to produce [32].

The energy conversion of the photovoltaic cell is described as a percentage of the maximum P_p_ power received from the incident P_i_ light energy; this is given by Equation (1). The filling factor (ff) is the calculation of the quality of a solar cell and is based on the short circuit current I_sc_ on the open-circuit voltage V_oc_. To attain the ff, the V_oc_ and I_sc_ are separated by the maximum power point [33].
(1)$$\eta =\frac{P_{p}}{P_{i}}=\frac{V_{P}I_{P}}{P_{i}}=\frac{{\mathrm{ffI}}_{\mathrm{sc}}V_{\mathrm{oc}}}{P_{i}}$$

Photovoltaic cells that are specifically tailored for the purpose of energy harvesting can be used in both indoor and outdoor environments. The light intensity in indoor environments is usually lower than that of the outdoor environment.

The power intensity produced by the Sun is far higher than that of an artificial source of light such as a halogen lamp, incandescent light bulb, or fluorescent tube. Therefore, to achieve the maximum feasible power, the spectral properties of solar light must be considered, because these spectral characteristics determine the operating range of each kind of light. The efficient performance of a photovoltaic cell, which depends on the material of which the cell is made, is higher for that reason in a particular wavelength range. Figure 2 illustrates a general model proposed in [28] for a photovoltaic harvesting system.

### 3.2. Thermoelectric Energy Harvesting

Thermoelectric harvesters are more suitable for environments with temperature gradients. The conversion of energy can be done efficiently using the temperature gradient between the material terminals, while the power is supplied by the heat flow. Due to the Carnot theorem and the low efficiencies of the material for thermoelectric devices, these decives do not provide enough energy, even when the heat flow is significant. Thus, the best thermoelectric materials are found in semiconductors that are heavily doped [33]. There are three effects through which thermoelectric devices operate:the Peltier effect, the Seebeck effect, and the Thomson effect. The Seeback effect is the essential thermoelectric power generation effect, which explains the situation in which a circuit consisting of two different materials can produce an electromotive force (EMF) or potential difference. Here, different temperatures are used for the maintenance of the junctions between the two materials. Thermoelectric devices have many advantages: they have no moving parts because they are in solid-state, they are highly reliable, they are acoustically silent, they are electrically safe, they are able to be orientated in any direction, they do not produce dust or other particles, and they emit no electric sound or flashing. In addition, they are small and light. The main drawback of this often-touted technology is the generally low conversion rate, which is from 5% to 10% at a normal power density of 60 μW/cm^2^. Such a drawback does not have a significant impact on the application, however, when other energy-harvesting techniques are considered [22]. Figure 3 displays the basic operation of the thermoelectric generator (TEG). In evaluating the overall output of TEG, as seen below, the Carnot efficiency is used [31].
(2)$$\eta =\frac{\left(T_{H}-T_{C}\right)}{T_{H}}$$
where *T_H_* and *T_C_* represent the hot and cold temperatures, respectively, in Kelvin.

### 3.3. Vibration Energy Harvesting

Another way through which energy can be harvested is vibration or mechanical movement. Figure 1 shows that there are three mechanisms (electromagnetic, and electrostatic, piezoelectric) that can be used for the conversion of vibrations into electrical energy [31].

#### 3.3.1. Electromagnetic Conversion of Vibration

Electromagnetic harvesters produce energy by means of the electromotive force that a varying magnetic flux induces through a conductive coil according to Faraday’s law [33]. A permanent magnet is used to generate the source of magnetic flux (B). The shift in the magnetic flux needed to cause a current in the coil is created by the motion of a seismic mass connected to either a magnet or a coil. The induction of the electromotive force (EMF) is induced between the ends of the conductor, while the conductor passes through a magnetic field. The proportion of induced voltage produced in the conductor (V) as shown in Equation (3) is proportional to the frequency of the circuit’s magnetic flux linkage (∅). The generator is a multi-turn (N) coil; thus, permanent magnets generate the magnet field.
(3)$$V=-N\frac{d\emptyset }{dt}$$

There are two possible cases:Linear vibration;Time-varying magnetic field, B.

With linear vibration, there is a movement between the coil and the magnet in the x-direction, meaning that the induced voltage of the coil and the movement velocity as shown in Equation (4) are expressed as a result of a gradient flux connection,. In the situation of the magnetic field varying in time (B), the flux density is constant across the area, A, and the coil, meaning that the induced voltage relies on the angle (α) between the coil area along with the axis of flux density, as outlined in Equation (5).
(4)$$V=-N\frac{d_{\emptyset }d_{x}}{d_{x}d_{t}}$$
(5)$$V=-NA\frac{d_{B}}{d_{t}}\sin\alpha$$

In order to extract power from the generator, a connection is established between the coil and load resistance *R_L_*. A magnetic field is generated by the current that is induced in the coil; furthermore, the magnet field is parallel to the magnets developed on the basis of the rule of electromagnetic induction by Faraday Lenz permanent magnets. The outcome of this electromagnetic induction is an electromotive force, *F_em_* which contradicts the generator motor by which the mechanical energy is converted into electricity. F_em_ is described in Equation (6); it is also proportional to current and speed.
(6)$$F_{em}=D_{em}\frac{dx}{dt}\;,D_{em}=\frac{1}{R_{L}+R_{c}+j\omega L_{c}}{\left(\frac{d\varphi }{dx}\right)}^{2}$$
where *D_em_* denotes the electromagnetic damping, *R_L_* is the load, *R_c_* is the coil resistance, *L_c_* is the coil inductance, and d/dx is the magnetic flux. Thus, *D_em_* and speed must be maximized by the generator design so that the maximum electrical power output can be obtained. If *D_em_* is increased, this means the flux connection gradient is maximized and the coil impedance is raised. The flux linkage gradient depends on the magnets’ strength, their relative positions on the coil and the direction of their movement, and on the area and number of spins for the coil.

#### 3.3.2. Electrostatic Conversion of Vibration

Electric harvesters [33,34,35,36] are made of variable condensers, the plates of which are electrically separated by air, a vacuum or insulator; the gap between the plates differs because of external mechanical vibration, thereby causing a change in the capacitance. For energy to be harvested here, the plates must be charged. Under such circumstances, the electrostatic forces in the device are opposed by the mechanical vibrations. Thus, in the event that a voltage V biases the capacitor, and if load circuitry is linear, electrical power is produced by the motion of the movable electrode. There are several advantages of electrostatic energy compared with other methods of harvesting vibrating energy. Some of these advantages include the wide range of tuning, less noise, high-quality factor Q, and limited size. Nevertheless, the energy produced by electrostatic harvesters is less than that produced by other kinetic harvesters, and because of their operational features, they have a restricted amount of functionality.

The mechanical action of the harvester is based on the main side of the models of the converter. The vibration source is represented by the voltage source, the mass is represented by the capacitor, the inductor is the spring, and the resistor reflects the parasitic damping. The secondary side contains the generator’s electrical elements, where the piezoelectric material or moving capacitor terminal capacitance is determined by the capacitor.

The mechanism of electrostatic conversion is dependent on a microelectromechanical system (MEMS) variable capacitor. A capacitor that was previously charged is placed in a system in a way that shifts the location of the conductors by mechanical movement, the aforementioned capasitor is made up of these conductors. Variability in the direction of the conductors allows the capacitance value to shift. Thus, there are changes in the energy stored in the capacitor, because mechanical energy is turned into electric energy. A capacitor can also be utilized to hold the transferred energy or it can be used as external load. However, because the capacitor needs to have been charged prior to its use, this type of system requires a different source of voltage [27].

#### 3.3.3. Piezoelectric Conversion of Vibration

In piezoelectric harvesters [33,34,37], mechanical elements such as membranes or beams are bent to enable energy generation. The resultant mechanical vibrations are transferred from tens to hundreds of Hertz at resonance frequencies. In a dynamic system, many factors, including an imbalance of weight and the wear or tear of the components, induce vibrations on a rigid body. Every system acts in a unique manner, which could be explained by the damping constant and natural frequency. A single degree of freedom lumped spring–mass mechanism explores the dynamic properties of the vibrating body correlated with energy harvesting. Therefore, the resonant inertial vibration harvester’s fundamental principles can be defined by utilizing the lumped model. The motion equation of the system is defined by D’Alembert’s theory regarding the energy balance equation provided by the differential, as shown in Equation (7).
(7)$$m\;\frac{d^{2}z}{dt^{2}}+D_{v}\frac{dz}{dt}+kz+F=-m\frac{d^{2}Z}{dt^{2}}$$
where m represents the seismic mass, the coefficient of *D_v_* is the viscous damping, *k* is the spring of stiffness, *F* is the power, and the place of equilibrium is *z(t)*. Since energy production is dependent on the relative movement process between the mass and the inertial frame, Equation (8) provides the standard steady-state solution for mass displacement:(8)$$z\left(t\right)=\frac{\omega^{2}}{\sqrt{{\left(\frac{k}{m}-\omega^{2}\right)}^{2}+{\left(\frac{D_{v}\omega }{m}\right)}^{2}}}Y\sin\left(\omega t-Ф\right)\;,\;Ф=arc\tan\left(\frac{D_{v}\omega }{k-\omega^{2}m}\right)$$
where *ω* is the frequency and *Y* sin (ωt − Ф) is the steady-state solution for z(t), with *Y (Y = A/ω*^2^) being the displacement of the amplitude and *Ф* the phase-shift. The maximum energy of a system is obtained when the excitation frequency is proportionate to the system’s natural frequency, *ω_n_*, given by Equation (9):(9)$$\omega_{n}=\sqrt{\frac{k}{m}}$$

The maximum power can be produced, and therefore the output power can be supplied by Equation (10), only when the equation is shifted at its normal frequency.
(10)$$P_{max}=\frac{mY^{2}\omega_{n}^{3}}{4\zeta_{T}}$$

The peak power is denoted by *ω_n_*, whereas the damping factor is denoted by *ζ_T_*. The magnitude of excitation vibrations and the influence of the frequency used should, therefore, be taken into account alongside the maximum displacement mass. An adequately high input acceleration causes an increase in the damping, which broadens the bandwidth; thus, a generator which is less responsive to fluctuations in arousal frequency is created. In the long run, environmental factors of temperature gradients may lead to changes in frequency. Additionally, a nonlinear behavior can be demonstrated by a device with a considerable amplitude of oscillations, thereby making it difficult to maintain the operation of the generator at resonance. For the power output to be maximized, the designing of the damping level and frequency of the generator must be done in a way that allows them to correspond to some explicit needs for the application. The amount of energy collected is proportional to the mass, which must be optimized according to the specific size restrictions. Only by examining the vibration spectrum is it possible to determine the most appropriate operating frequency for this generator size, configuration constraint, and peak displacement.

One of the main benefits of the process of piezoelectric energy harvesting is the fact that it could utilize a wide variety of sources of vibration. Further power could be produced by increasing the resonant frequency of the unit as the frequency of vibrations increases. The typical effectiveness depending on the nature of crystal utilized is 0.5% for polyvinyl fluoride (PVDF) and up to 20% for lead. Figure 4 shows the schematic representation of the concept of a piezoelectric energy harvesting system [38].

### 3.4. Radio Frequency (RF) Energy Harvesting

There is a constant availability of RF signals [31,39,40], particularly in heavily populated mega-cities. Thus, WBANs can be charged wirelessly when RF signals are harvested from the environment. Furthermore, environmental factors such as weather and temperature do not affect RF scavenging, and this is significant for WBAN apps. The different sources of ambient RF are as follows:AM radio band (550 kHz–1605 kHz);FM radio band (87.5 MHz–108 MHz);TV band (41MHz–950 MHz);GSM band (0.85–0.90 GHz, 1.8–1.9 GHz);Code Division Multiple Access (CDMA) band(450 MHz–2100 MHz);3G band (1.8–2.5 GHz);4G band (2–8 GHz);ISM band (2.4 GHz);WIFI band (2.45 GHz–5.8 GHz).

An antenna is utilized to collect radio signals from the atmosphere in RF spectrum scavenging. The system has impedance matching circuits to fit the remaining circuit to the antenna, thereby preventing the occurrence of reflections. After the RF signals have been captured, they are then converted to DC through the use of different available approaches that are reliant on applications. Then, the modified signal is then redirected in wireless body area network (WBAN) applications to the energy management device. The amount of power that RF energy scavenging systems can harvest depends on the antenna gain, the source power, and the distance between source and antenna. Ambient RF scavenging’s density of output is between 0.0002–1 µW/cm^2^. In comparison with other methods of scavenging, this range of power output is comparatively low. The distance sensitivity is another key point that should be considered in RF scavenging. In Table 2, the energy levels, efficiencies, and limitations of the different sources of RF are presented. Nevertheless, the ability of the RF scavenging method to always capture RF signals in both outdoor and indoor environments makes it attractive for WBANs. Furthermore, the RF scavenging approach is increasingly important for future implementations due to the possibility of antenna integration as well as the development of antenna design techniques.

### 3.5. Hybrid Energy Harvesting

Since the ambient harvestable sources are not continuously available, it is unwise to rely on only one source, as this limits the reliability and functioning capacity of the node of the sensor may completely stop the functionality of a node. A sensor node can harvest energy from multiple revolving sources, thereby permitting the optimal utilization of the ambient energy by the sensor node. A sensor node can be equipped with several harvesting modules, enabling it to harvest energy from multiple sources (multi-sources harvesting). With this system, wind and solar energy can be harvested; maximum power point tracking (MPPT), which consists of two 22 F super-capacitors and a 70 mAh Li-polymer battery that serves as a subordinate buffer, is also used. The solar panel device has a maximum current and voltage output of 3.75 and 2.5, respectively, equivalent to the Helimote, between 4.0 V and 100 mA. An output power of 400 mW can be generated by the panel [41].

Because hybrid scavenging is a recent development, high-output production and the increased reliability of several hybrid systems is anticipated in the future. In addition, the majority of the studies conducted in this area are centered on the integration of all approaches to harvesting into one circuit for the sake of facilitating the extraction of the maximum obtainable energy, eliminating the use of the batteries in WBANs or increasing their communication range and capability [31]. The system combines a thermoelectric generator (TEG) as well as vibration-based electromagnetic (EM), and piezoelectric transducer (PZT) harvesters. Figure 5 shows the structure of the harvester. In addition, Table 3 shows the basic characteristics of the method of scavenging, and then the methods are compared in terms of their disadvantages and advantages; each technique is also briefly explained.

## 4. Rectenna

The rectenna, which is able to transform RF energy into DC, is commonly utilized for wireless power transfer (WPT) systems and also for RF-EH systems. The rectenna, therefore, is generally responsible for determining the overall RF to DC power conversion efficiency (PCE). A high PCE can be achieved through the use of two approaches. Nonetheless, the commonly employed method is to absorb the maximum power and transmit it to the revising circuit. The use of broadband antennas, large array antennas, etc. can be applied to achieve this, at the disadvantage of broad sizes. A low-pass filter between the rectifier and antenna is used in the second method. Additionally, harmonic antennas of rejection could be designed to ensure that the nonlinear elements which are present in the rectifier do not re-radiate signals [39,42]. The rectenna is basically made up of a matching circuit, harmonic suppression filter, antenna, voltage multiplier, and load, as shown in Figure 6. Ambient/dedicated RF signals can be received by the antenna, and the use of the matching circuit is employed to match the necessary circuit’s antenna impedance to reach high performance. Furthermore, the matching circuit can be used as a filter to eliminate re-radiated signals. The AC signals are converted to DC using the rectifier, and the output voltage is determined by the number of multiplier circuit stages. The use of voltage multipliers is required so that the output DC can be boosted because the DC output voltage amplitude is greater than the obtained RF signal amplitude. The major challenge associated with the design of the rectenna is its power conversion efficiency (PCE). The rectifier’s conversion efficiency is measured by the PCE. The PCE is used to measure the rectenna’s effectiveness to convert the received RF energy into DC current [39].

Equation (11) shows the efficiency of conversion of the RF to DC rectenna:(11)$$\eta_{Rf-DC}=\frac{P_{Dc}}{P_{Rf}}$$
where P_DC_ is the output DC power and P_Rf_ is the input RF power. There is a variation in the rectifier’s input depending on the strength and frequency of the incident wave, which causes a difference in the diode impedance. As a result of this variation, impedance mismatch occurs, thereby creating a decrease in the PCE. Several factors, including the following, influence the conversion efficiency of the RF-EH systems:Unpredictable incident RF power availability;Losses for every component;Minimal associated circuit responsiveness;Restricted overall power of radiation;The distance between transmitter and receiver;High non-linear dependence of the output voltage on the input at low input powers;Variation in the antenna output impedance because of variation in the incident power and frequency leading to mismatch losses;A trade-off between bandwidth and efficiency due to miniaturization.

In research conducted by Okba, Takacs and Aubert [44], a rectenna was presented and implemented with the best trade-off between compactness and high RF-to-DC efficiency for low incident power density levels (<5 μW/cm^2^). In the study by Palazzi et al. [45], a new compact ultra-weight multiband RF energy harvester was introduced that was produced on a paper substrate. A new rectenna design was proposed by Cambero et al. [46] for solar and RF energy harvesting. Almohaimeed, Amaya, Lima, and Yagoub [47] proposed and implemented an adaptive rectifier concept to address the problem of the early breakdown that occurs in conventional rectifiers while demonstrating a high level of efficiency over a wide range of RF input power levels. A new rectenna system was proposed by Chen and You [48], and the proposed rectenna was able to achieve multidirectional reception and scalability at the same time. In their work, the power generated by the antennas with varying half-power beamwidths (HPBWs) was evaluated through the use of the ray-tracing technique. A novel, compact, ultra-weight RF energy harvester was proposed by Eid et al. [49]; the proposed harvester was produced on a flexible substrate. The proposed rectenna was designed to operate over the 2.4 GHz industrial, scientific, and medical radio band (ISM). In the work of Bakogianni and Koulouridis [50], an arm-implantable rectenna was proposed; a lightweight inverted planar F-antenna (PIFA) and a rectifier were also fitted with a planned rectenna. The researchers suggested the use of the rectenna in the medical device band (401–406 MHz) and in industrial, scientific research and medical (ISM) bands (902.8–928 MHz) to provide wireless data telemetry and transmit power. A novel, low-profile, implantable planar dipole was introduced by Lesnik, Verhovski, Mizrachi, Milgrom, and Haridim [51], in which a folded meander section was included in the proposed dipole. The work made two new contributions: firstly, a specially tailored folded meander section was used; secondly, the replacement of only part of each otherwise straight arm by the meander section was performed. A summarize of different kinds of rectenna in state of art is illustrated in Table 4.

## 5. Antenna

As defined in Webster’s Dictionary, an antenna is “a usually metallic device (as a rod or wire) for radiating or receiving radio waves.” According to the standard definition given by the IEEE Standard Definitions of Terms for Antennas (IEEE Std 145 1983)∗, an antenna is a trough in which radio waves are radiated or received. The antenna is one of the major parts of the WLAN network. With a properly designed antenna, system requirements can be relaxed, while the general performance of the system is improved. A classic example of this is Television, for which a high-performance antenna can be used to improve the general performance of broadcast reception. Just as the eyes are to humans, so is the antenna to communication systems [52].

An antenna is a key component of RF energy harvesting systems, and there are different kinds of antennas [53]. The classification of antennas is based on antenna gain, polarization, frequency band, radiation pattern, application area, and physical dimensions, etc [54]. For example, for different frequency bands such as very high frequency (VHF) and ultra high frequency (UHF), the use of specific antennas may be of great benefit. Some of the popular types of antennas include loop, horn, dipole, aperture, array, microstrip, and log-periodic antennas. The conversion efficiency from RF to the electrical signal is calculated by antenna gain in a particular direction. Antenna with high gain is most desirable as they increase the conversion performance as well as the amount of energy they harvest. Antenna gains have a large cost. The antenna radiation can be either isotropic or directional. If the RF signal source is established, the amount of energy obtained could be improved with the aid of the directional antenna. However, if the location is unknown, an isotropic antenna is employed. An electric field’s orientation at an observation point is defined by the antenna polarization. An increase occurs in the conversion efficiency when the same polarization is possessed by both the transmitting and receiving antenna. There are four types of polarizations, including vertical, horizontal, elliptical, and circular [53].

The whole electromagnetic energy distribution in the region around the harvesting antenna exists in different spectral bands. As earlier noted, the spectral measurements are used to determine a promising frequency band or bands. The choice of the design of band (whether broadband, multiband, or single band) to be used in RF energy harvesting is determined based on the measurement results. The process of designing and manufacturing a single band antenna is simple, but the energy harvested by the single band antenna is less than that of the multiband antennas [53]. However, the output DC voltage of the RF power harvester increases when the power harvester’s circuit is configured for a multi-frequency band design [55]. The most suitable kind of antenna for harvesting power from a wide frequency band is the broadband antenna, but a decrease occurs in its antenna gain when it is away from the center frequency [53]. Besides, it is challenging to achieve the impedance matching across a wideband. When there is need to harvest more power, the use of multiple antennas is employed, and the RF-DC conversion efficiency is enhanced by this more power that is harvested [56,57], but when the multiplier stages and the number of the antenna is increased, an increase will also occur in the efficiency, as well as the overall circuit size [58].

It must be noticed that efficiencies are possible to raise the harvested energy, but any increase in the harvested energy does not guarantee increased efficiency. For example, the operation of two similar antennas working in an energy harvesting system with the same frequency band will increase the amount of the harvested energy. Still, the system’s output stays the same in this case [48]. Yang, Sun, & Guo [59] developed a low-cost, series-fed dual-CP antenna printed on a single-layer PCB for MM-wave applications. In the work of Cheng, Wang, & Liu [60], a 94 GHz substrate integrated waveguide (SIW) parallel-plate long-slot array antenna was presented. With the proposed antenna, dual-circular-polarization (CP) low sidelobe level (SLL) beams can be generated from a single radiating aperture. Zhao & Luk [61] proposed a dual circularly polarized high gain scalable antenna array for the 60 GHz applications. In the study carried out by Park & Park [62] Left-hand and right-hand circularly polarized (LHCP and RHCP) substrate integrated waveguide (SIW) antenna arrays at the 28-GHz band were proposed for millimeter-wave (mm-wave) applications. A new low-profile dual Cp-planar aperture antenna was proposed by Zhu, Liao, Yang, Li, & Xue [63]. The proposed antenna is characterized by wide bandwidth, high gain, and excellent axial ratio (AR) for 60 GHz applications. A novel scheme for building a dual-polarization was developed based on the proposed antenna. A wideband multiple-microstrip dipole antenna with dual polarization was proposed by Zhou, Wei, Tang, & Yin [64]. A novel broadband flush-mountable dual-polarized antenna was proposed by Cui, Niu, Qin, & Li [65]. The proposed antenna has very high isolation. Lian, Wang, Yin, Wu, & Song [66] proposed a low-profile microstrip-fed dual-polarized stepped impedance (SI) slot antenna element. In the study of Liu, Wang, Wang, & Jia [67], the differential feeding technique was combined, and a dual-polarized slot antenna was proposed. The proposed antenna is characterized by high isolation, a consistent pattern of radiation, and XPD, these characteristics are desirable in base station applications. Zhou, Wong, & Yeung [68] introduced a wireless cross-slot antenna. The inductor loadings are given in the antenna for double polarization. The authors investigated the effects which the ground size has on the impedance. A compact size can be achieved by using a very small ground that has the same hypotenuse length as the slot. Table 5 below presents a summary of the comparison of previous studies on antennas.

## 6. Types of Antennas

Although there are different kinds of antennas, only a few of them are explained here as Figure 7.

### 6.1. Dipole Antenna

The dipole antenna is the most common antenna used in telecoms and consists of two metallic bars in series and a feeder. The total length of the two metal bars is equal to half of the wavelength of the RF signal or one-quarter of the wavelength of each metal bar, which defines the antenna’s resonant frequency. Figure 8a shows a complete dipole antenna comparable circuit. The conversion of the impedance into resistance occurs if the signal is received by the antenna at the resonant frequency. Figure 8b [69,70] shows a simple equivalent antenna circuit functioning at a resonant frequency.

### 6.2. Microstrip Patch Antenna

Figure 9 indicates that the microstrip antenna consists of a radiation patch, a ground plane, a dielectric substrate, and a nourishing point that could be used in a variety of ways. The radiator patch dimensions thickness (h), width (W_P_), and length (Lp) determine the bandwidth, resonant frequency, and efficiency of the microstrip antenna [71]. The polarization, frequency band number, antenna gain, and bandwidth are all determined by the shape of the radiator patch (rectangular [72,73,74], circular [75,76], square [77,78], and others [79,80,81]), as well as the feeding technique (position, type, and number of feeds). Plans of various types of techniques for the bandwidth improvement of microstrip topologies have been made: (I) patch modifications generating complex designs to mix various resonant frequency bands [73,75], (II) the addition of radiator patch slots [72,75,79], (III) the insertion to the radiator patch layer of parasitic components [72,79], and (IV) adjustments in the feeding technologies [72,82,83].

For microwave frequencies, particularly on the 2.4 GHz band and above, the patch antenna is useful. This antenna consists of one side of a printed circuit board of a plated geometric design and a ground plane beyond the parched radiation dimensions. The most common designs for this antenna include circular and rectangular forms. However, other designs are sometimes employed, such as trapezoid. The board is exposed to excessive radiation. A square half-wave patch antenna is required for a range of 7 to 8 dB [84].

A rectangular patch antenna is shown in Figure 9. Dimension L is about half a wavelength, which is determined to be half of the free space wavelength (λ) separated by the square root of the effective dielectric constant (ε) of the substance of the board. The fringing influence of the radiation from the two opposing patch edges, which are L apart, and from the ground plane, is considerably less than half a wavelength. Radiation does not emanate from the two other regions in as much as they feed in positioned on the centreline. A microstrip feeder can be observed in Figure 9, and because it is attached to the patch on the board alongside further element traces on the same side, it is preferable to other designs. The width of the patch determines the impedance at the feed point. The microstrip transforms the impedance into the best load (for the transmitter) or source (for the receiver) impedance. The feed point can be moved from the edge of the centreline to the middle of the board to directly fit a transmission line for the impedance of the feed point. In this way, a direct connection can be established between a 50 ohm coax transmission line and the underside of the patch antenna. In this case, the conductor passes to the feed point, whereas the shield is connected to the ground plane [84,85].

### 6.3. Horn Antenna

The horn antenna depicts a transfer or equivalent component from the guided mode within the waveguide to the non-guided (free space) mode of the waveguide. The horn antenna reduces the reflections as a matching component, thereby resulting in a reduced standing wave ratio (SWR). The horn antennas have three types, as shown in Figure 10: (a) the sectoral E-plane horn (flared only in the E-plane direction, (b) the sector-based H-plane horn (flared only in the H-plane direction), and (c) the pyramidal horn antenna (flared both in E-plane and H-plane) [86]. The assumption here is that the flare of the horns is linear even though the formation of some horns is carried out through other kinds of flare. An exponential flare is an example of the latter. The horn antenna is mounted on a waveguide, which is almost continuously excited in single-mode operations. In other words, the operating frequency of the waveguide is above the TE_10_ mode cut-off frequency, yet below the cut-off frequency of the next highest mode. The horn is typically utilized for communication, dishes, large-scale radio astronomy, and satellite surveillance. This is because the horn is simple to construct, is easily excited, has large gain, and performs well overall. The end of the waveguide can be flared in different ways to enable the creation of a variety of horns. For example, a sectorial horn can be created by flaring the E and H planes. In the same vein, a pyramidal horn can be formed when the waveguide is flared in both dimensions, or a conical horn can be produced by flaring a circular waveguide [87,88].

### 6.4. Loop Antenna

The popularity of the loop antenna lies in its application in hand-held transmitters, particularly since it can be made on a circuit of a small board. It is also utilized widely because the influence of the surrounding conducting objects is less than other low-resonant antennas. The biggest limitation of this antenna is that it is highly ineffective. The current of a loop antenna with measurements that are less than a wavelength smaller than 0.1 λ is constant [84]. The equation below is an expression of the radiation field:(12)$$E\left(\theta \right)=\frac{120\pi^{2}.I.N.A}{r\lambda^{2}}cos\theta$$
The loop zone is indicated by *A*, where *I* represents its current, *r* is the distance, the angle from the plane of the loop is *θ*, and *N* is the number of loops in the case of multiple loops. The resistance to radiation stems from the following expression:(13)$$R_{r}=320\pi^{2}.\frac{{\left(A.N\right)}^{2}}{\lambda^{4}}$$

In handheld remote transmitters operating at low ultra high frequency (UHF) frequencies, loop antennas are usually employed. These are typically defined by a radiation resistance below 10% and an efficiency lower than 10%. Here, the use of a capacitor is employed across the loop terminals so that parallel resonance can be created to enable the matching of the transmitter output stage to the low antenna resistance. Even though it may seem that, in this approach, an increase occurs in the efficiency and radiation resistance when the area of the loop or number of turns increase, this approach offers limited possibilities. The loop inductance increases as the turns or area increases, thereby needing a smaller resonating capacitance value. When the resonant capacitance is as low as various picofarads, the approach reaches its limit. The poor performance of this approach renders it unattractive for use in UHF receivers for a short distance. Pager receivers are unsual since their use of low data levels is highly sensitive and thus balances poor antenna performance. However, the loop antenna is useful in the sense that it does not need a ground plane [84,89].

### 6.5. Helical Antenna

In comparison with the loop antenna, better results can be provided by a helical antenna in terms of radiation efficiency. In contrast with a dipole or a quarter-wave ground plane, the antenna is relatively small.

The helical antenna is made up of a spring with a diameter far smaller than a wavelength or by a winding wire in a cylindrical shape, as shown in Figure 11 [84,90]. This helical winding produces an apparent axial speed along with the coil, which lies below the propagation speed along a straight wire, which is almost light speed in a vacuum.

A quarter-wave will, therefore, be far shorter on the helical source than on a straight wire. The length is sufficient for the antenna, but the resistance to radiation is smaller; thus, the efficiency of the regular quarter-wave antenna is lower. The wire length in the region of half a wavelength induces the resonance of the helical antenna. The impedance of a transmitter or receiver is quite easy to match. The polarization of a helical antenna is elliptic because its radiation has both horizontal and vertical components. Nevertheless, for the popularly used form factors, where the length of the antenna is much more than its diameter, polarization is vertical.For the predictable and optimal performance of the helical antenna, it must possess a good ground plane. In hand-held devices, the body and arms of the user serve as a counterpoise, and as such, the design of the antenna should be done in a manner that is appropriate for this configuration [84].

## 7. RF System of Harvesting Power Conversion

The overall block diagram for an RF system converting harvested power is demonstrated in Figure 12. The system generally aims to optimize the voltage that approaches the RF–DC power conversion system to stabilize the DC power output voltage at a low power level. The circuit comprises an antenna, a voltage regulator, an impedance matching network, a multi-stage rectifier circuit, and a load capacitor. The antenna takes charge of the power radiated by the RF waves. Maximum power transmission is assured in the system with an impedance matching network. The RF signal is converted to DC voltage using the multi-stage rectifier circuit. The fluctuation of DC power, which occurs as a result of rectification, is eliminated with the voltage regulator circuit. Lastly, the load capacitor is used for the storage of the DC voltage. Figure 12 demonstrates that two cascading circuits, including an RF-to-DC rectification mechanism and a DC-to-DC (regulator) conversion method, specify the effectiveness of a power harvesting system. It is important to identify the component that causes losses in the rectifier so that highly efficient RF power harvesting can be achieved; this can also be achieved by designing an energy-efficient regulator [91,92].

### 7.1. Impedance Matching Network

It is essential to create an adjustable network between the receiving antenna and the antenna rectifier to suit the impedance between the two, which could also minimize the transmission loss and increase the voltage gain [83]. Figure 13 shows an illustration of an antenna and rectifier equivalent circuit.

Capacitors and inductors can simply build the impedance matching network. Capacitive (C_rec_) and resistive (R_rec_) components are the elements that are possessed by the Z_rec_ rectifier’s input impedance.
(14)$$Z_{\mathrm{rec}}=R_{\mathrm{rec}}\|\frac{1}{{\mathrm{sC}}_{\mathrm{rec}}}$$

It is noticeable in Figure 13 that the sequence of R_rec_ and C_rec_ networks can be acquired at resonance by the conversion of resistive and capacitive components of the rectifier.

At resonance [94],
(15)$$Z_{s}=R_{s}$$
the power dispersed over the antenna is equivalent to that dispersed over the resistive component R_sr_ in the rectifier.

Equation (16) is an expression of the correlation between the voltages across R_s_ and R_sr_:(16)$$V_{IN}=V_{s}Q_{sr}\sqrt{\frac{R_{sr}}{R_{s}}}$$
where *V_s_* and *V_IN_* are the amplified input voltage and the voltage at the input of the rectifier, respectively. Thus, the presence of the matching network can provide passive input voltage amplification to the receiving voltage at the antenna. With an increase in the amplitude of the voltage which reaches the input of the rectifier, the output voltage of the rectifier also increases and, therefore, the overall power conversion efficiency of the system is increased.

### 7.2. Rectifier Circuit

The rectifier circuit is one of the most essential components of the rectenna RF energy harvesting system [18]. A rectification circuit is considered to be a non-linear circuit, which implies that the diode correlation current and voltage (IV) are not linear, which indicates that the input voltage is raised linearly and that the input voltage is “turned on” when the required input voltage is reached. This could be used for different applications of RF systems, such as oscillators, mixers and rectifiers, and mixers [95]. There is a conceivable nonlinearity.

A perfect diode acts simply as a switch (which is also non-linear) that is closed when the alternating current (AC) signal is in one polarity and is open when the AC signal is in the other polarity. The classic non-linear model [85] of the current on a diode can be written as
(17)$$I\left(V\right)=I_{s}\left(e^{\alpha V}-1\right)$$
where I_s_ is the saturation current (typically between 10^−^^6^–10^−^^15^ A), α is equivalent to q/nKT, where q is the charge of an electron (1.602 × 10^−^^19^ C), n is the ideality factor (between 1 and 2), k is Boltzmann’s constant (1.38 × 10^−^^23^ m^2^ kgs^−^^2^ k^−^^1^), T is the temperature (290 K for room temperature), and V is the voltage. Thus, a typical value of α is roughly 1/(25 mV) [80]. An ideal IV plot for a generic diode using Equation (17) as well as simulated and measured IV curves for an SMS7630 diode are shown in Figure 13. Since the exponential term in Equation (17) is quite large, it is often simplified as
(18)$$I\left(V\right)=I_{s}\left(e^{\alpha V}\right)$$
which illustrates the current as a non-linear function of the voltage. The non-linearity of the diode is responsible for creating DC as well as the harmonic components. This could be because, in Equation (18), the exponential may be represented as an extension of the Taylor series [95] in the following form:(19)$$e^{x}=\overset{\infty }{\underset{n=0}{\sum }}\frac{x^{n}}{x!}=1+x+\frac{x^{2}}{2}+\frac{x^{3}}{3!}+\cdots$$
and Equation (18) therefore becomes
(20)$$I_{\left(V\right)}=I_{\left(s\right)}\left(1+\left(\alpha V\right)+\frac{{\left(\alpha V\right)}^{2}}{2}+\frac{{\left(\alpha V\right)}^{3}}{3!}+\cdots \right)$$

This shows the creation of a DC component alongside the original signal and higher harmonics. A close look at the response of the diode in the frequency domain, as contained in Figure 14c shows this phenomenon. The spectral response and the time domain of a time-varying signal that has been applied to a diode are shown in Figure 14. The use of the Keysight Advanced Design System (ADS) was employed for all simulations of the rectifier [95].

Based on this vantage point, the RF-to-DC rectification can be expressed as follows:(21)$$\eta =\frac{P_{DC,out}}{P_{RF},in}$$
where *P_RF_*, *_in_* is the quantity of power in the frequency that is fundamental, and *P_DC, out_* is the power in the output DC signal [96].

### 7.3. Technologies of Energy Storage

The construction of an energy storage system may be one of the most challenging tasks in the domain of energy harvesting. In most cases, energy is stored in common storage systems such as an electric double layer capacitor (EDLC) and batteries. Each of these storage options is accompanied by some advantages and restrictions. Carbon is used as an electrode in EDLC, and charging is stored by aqueous or non-aqueous electrolytes in the electric field of the interface. EDLC charging is simply physical and not a chemical reaction, and as such it is incredibly reversible, which leads to increased hours of life, shelf life, and a product that is free of maintenance. A few benefits can result from using EDLC for storage [97], such as the following:Unlimited charge cycle life;High density of power;Thermal heat during discharge is absent;No risk of overcharging;Effect-free under deep discharges;Longer lifetime;A functional temperature range between −50 °C and 85 °C.

Intrinsic leakage is associated with supercapacitors due to the parasitic paths that are present in the external circuitry, thereby making them unusable for long-term energy storage [85]. In comparison with supercapacitors, batteries are comparatively advanced technologies that possess higher power density. In EH applications, the use of different kinds of rechargeable batteries is employed, including nickel metal hydride (NiMH), sealed lead acid (SLA), nickel cadmium (NiCd), and llithium-based (Li) batteries. Among these batteries, the least used ones are the NiCD and SLA batteries due to their temporary loss of capacity and comparatively low energy density. The problems associated with the aforementioned batteries occur as a result of their shallow discharge cycles, also known as the memory effect. There are several benefits and limitations that should be considered when making a choice between Li and NiMH batteries; for instance, the lifespan of Li batteries is longer than that of the NiMH, and their efficiency is also higher than that of the NiMH. In addition, the rate of self-discharge in Li is lower than that of NiMH batteries. Nevertheless, they are more costly than the NiMH, even after their increased lifecycle is accounted for. Furthermore, Li batteries need a charging circuit, which is significantly complicated. Despite the fact that high energy densities are not offered by batteries, the absence of a high power density makes them inappropriate for certain applications, such as sensor networks, that require instant power transmissions. Meanwhile, supercapacitors cannot be independently used as an energy storage unit because of their low energy densities [97].

## 8. Conclusions

Radio frequency energy signals are now ubiquitous, and they have indispensable features for RF energy harvesting systems. Compared to other energy sources such as thermal gradients, solar, and mechanical vibrations, the power density of RF energy is relatively low, but it is sufficient to charge some sensors or devices of harvested energy. Antennas are one of the essential parts of the developed energy harvesting system, and the performance of those antennas, such as their efficiency, gain and radiation patterns, is critical to improve overall performance. For this reason, it very important to select suitable antennas based on their intended applications, such as in RFID, 5G, WSN, etc. For instance, IoT-based sensors require a low-profile antenna with wide bandwidth and high gain to be used. However, antenna arrays with omnidirectional radiation patterns are dedicated to ambient RFEH systems, so signals can be received from all directions, improving the PCE of the system. Besides this, the backside radiation of antennas should be very low, as exemplified by wearable applications. An overview of energy harvesting is presented in this article with explanations of several modeling requirements such as the power supply converter architecture, additional storage system (capacitors or batteries), control systems which can manage and correlate the energy availability statistics over time, and the electronic load consumption patterns of the target application.

## Figures and Tables

**Figure 1 sensors-20-02772-f001:**
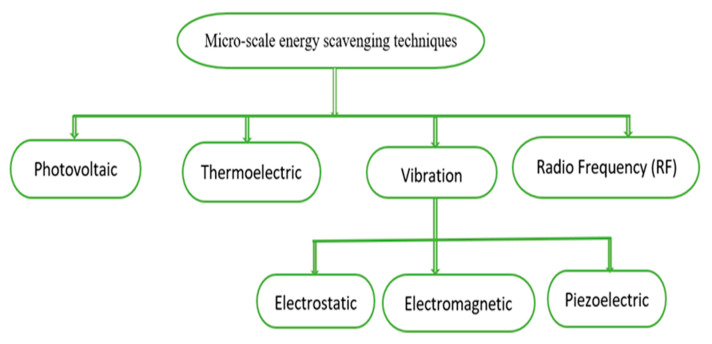
Micro-scale energy scavenging techniques. Reproduced from [31].

**Figure 2 sensors-20-02772-f002:**
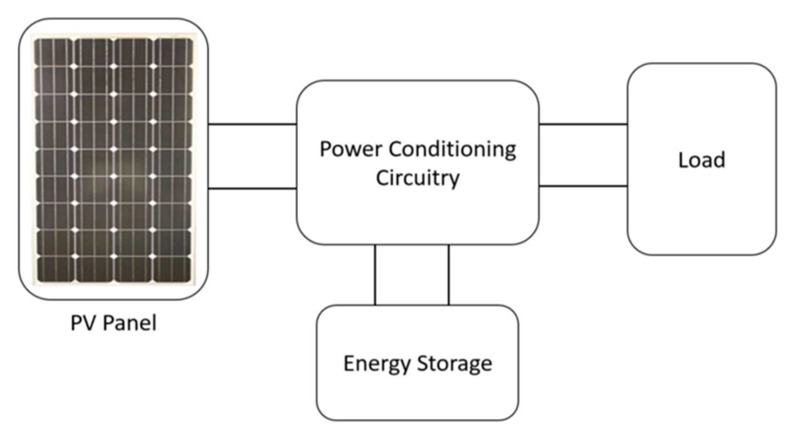
Photovoltaic energy harvesting system. Reproduced from [31].

**Figure 3 sensors-20-02772-f003:**
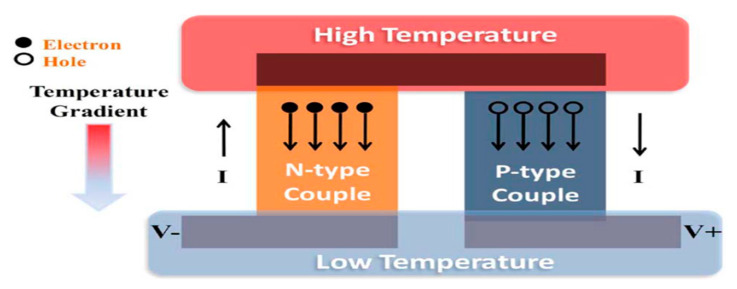
The basic operation of a thermoelectric generator (TEG). Reproduced from [31].

**Figure 4 sensors-20-02772-f004:**
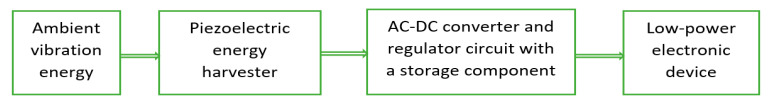
Schematic representation of the concept of a piezoelectric energy harvesting system. Reproduced from [38].

**Figure 5 sensors-20-02772-f005:**
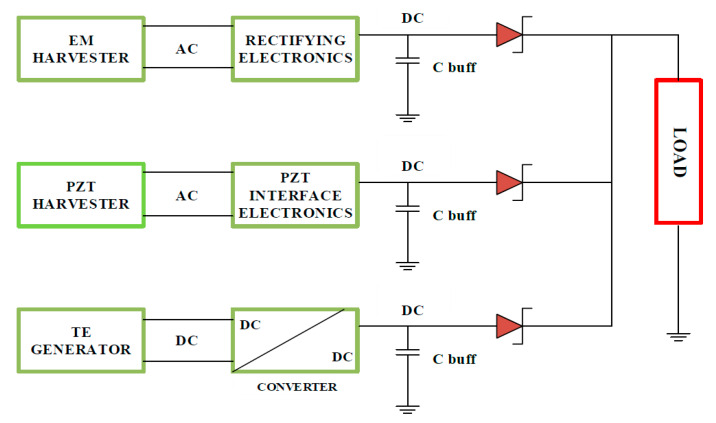
Hybrid harvester structure with triple sources. Reproduced from [31]. TE: thermoelectric; EM: vibration-based electromagnetic; PZT: piezoelectric transducer.

**Figure 6 sensors-20-02772-f006:**
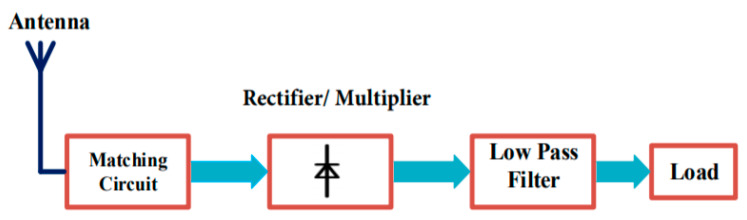
Block diagram of an radio frequency enerby harvesting (RF-EH) circuit [43].

**Figure 7 sensors-20-02772-f007:**
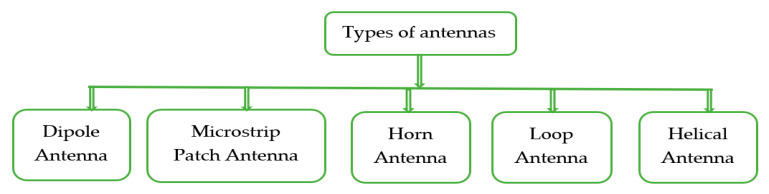
Types of antennas included in this review.

**Figure 8 sensors-20-02772-f008:**
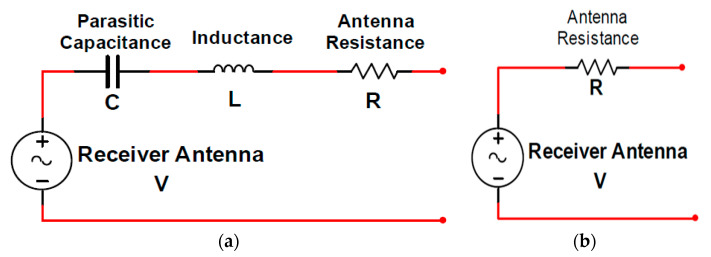
(**a**) Full dipole antenna equivalent circuit, (**b**) simple dipole antenna equivalent circuit for operation at resonant frequency. Reproduced from [69].

**Figure 9 sensors-20-02772-f009:**
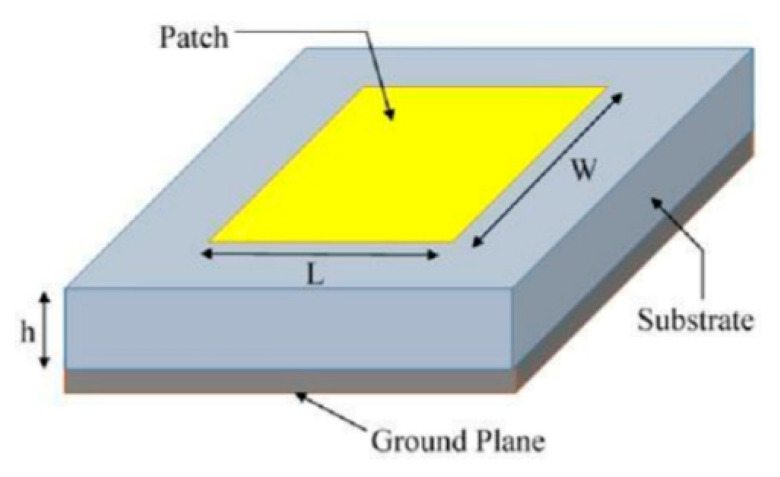
Basic structure of the rectangular microstrip patch antenna. Reproduced from [81].

**Figure 10 sensors-20-02772-f010:**
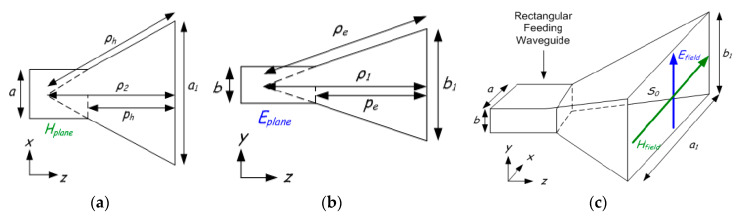
(**a**) H-plane, (**b**) E-plane and, (**c**) pyramidal horn. Reproduced from [86].

**Figure 11 sensors-20-02772-f011:**
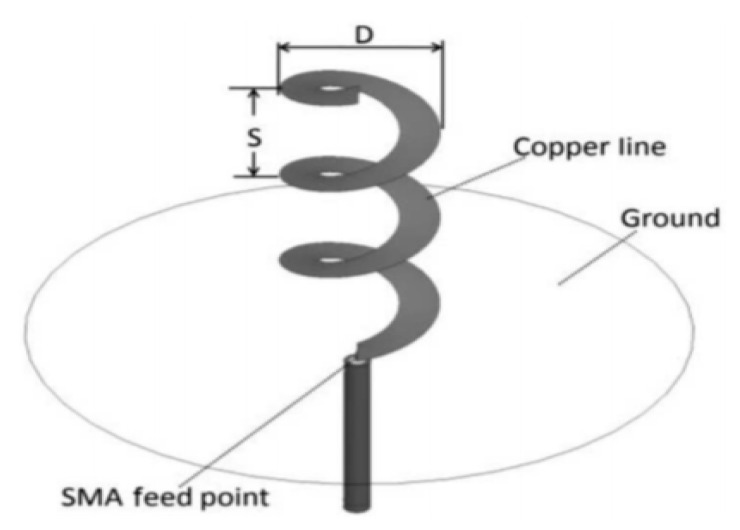
Geometry of a standard helical antenna. Reproduced from [90]. SMA (Sub-Miniature version A).

**Figure 12 sensors-20-02772-f012:**
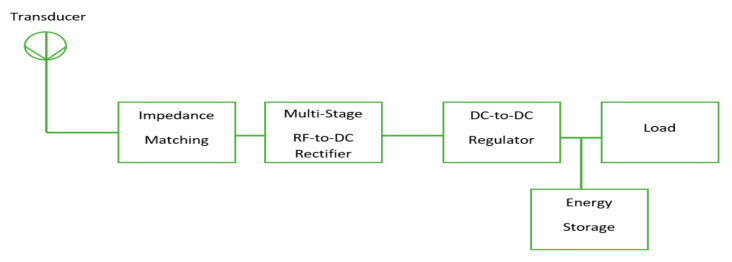
General block diagram of an RF power conversion system.

**Figure 13 sensors-20-02772-f013:**
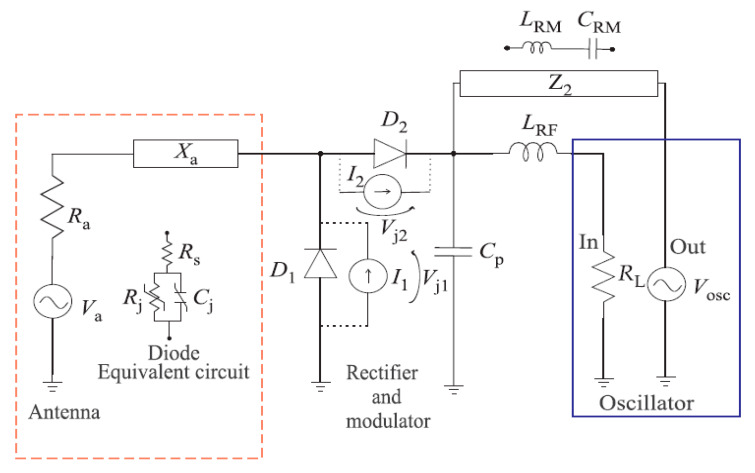
Equivalent circuit representation for an antenna and rectifier. Reproduced from [93].

**Figure 14 sensors-20-02772-f014:**
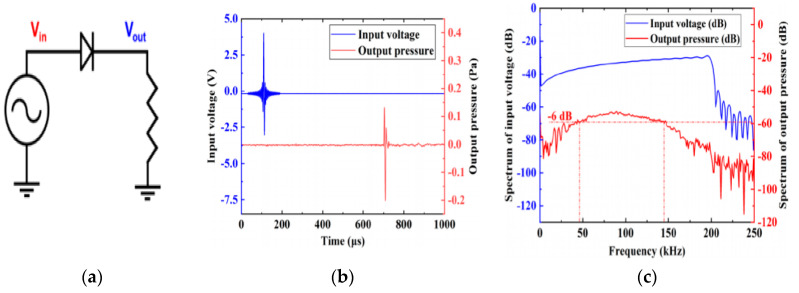
Frequency and time-dependent voltages of input and output: (**a**) diode schematic, (**b**) time domain view, (**c**) frequency domain view [95].

**Table 1 sensors-20-02772-t001:** Summary of previous studies on energy harvesting.

Ref	Antenna Type	Technique	Freq (GHz)	Eff. (%)	Gain (dBi)	Input Power (dBm)	Load (kΩ)	Dimension cm × cm
[22]	Tree-like antenna patch	--	2.45/3.5	60	--	0	0.9-20	35 × 37
[23]	--	Synchronous electric charge extraction (SECE)	6.5	79	--	--	--	50 × 100
[24]	Slotted antenna	--	2−31	70	2.5	5	2−20	35 × 50
[25]	Horn antenna	Full wave electromagnetic simulator and harmonic balance simulator	5.0	48	15	0	0.5−1.1	20 × 20
[26]	Horn antenna	Polarization diversity	0.94−1.84	42	3−4	−20	7	240 × 240
[27]	Coplanar waveguide (CPW)-fed rectangular antenna with slots	Tuning stub technique	2.45	68	5.6	5	5	13 × 18
[28]	Annular ring slot and periodic antenna array	Complementary metal oxide semiconductor (CMOS) technology	0.9 2.025 2.360	59	1 2.64–0.19	−10	2	70 × 66
[29]	Microstrip antenna	Multilayer substrate technology	1.85–1.93 2.0–2.1	53.6	3.8–9.3	0	10	20 × 07
[30]	Helical antenna	Inductive coupling and resonant coupling	0.424 0.441	66.8 76	2	−3 0	4.7	41 × 5.2

**Table 2 sensors-20-02772-t002:** Radio frequency (RF) energy scavenging comparison for different frequencies [40].

Source	Efficiency	Energy Level	Limitation
**RF-GSM**	Low	mW	0–100 m
**RF-TV**	Low	μW	0–4 km
**RF-WIFI**	Low	ոW–μW	0–10 m
**RF-AM**	Low	μW–mW	0–20 km

**Table 3 sensors-20-02772-t003:** Energy harvesting methods and their characteristics [31].

	PV Solar	Thermoelectric	Piezoelectric Vibration	Electromagnetic Vibration	Ambient RF
Power density	Outdoor: 100 Mw/$cm^{2}$ Indoor: $<100\mu W/cm^{2}$	50–100 $\mu W/cm^{2}$ per °C	10–200 $\mu W/cm^{3}$	1–2 $\mu W/cm^{3}$	0.0002–1$\;\mu W/cm^{2}$
Output voltage	0.5 $V_{max}$	10–100 Mw	10–20 V (open ckt)	Few 100 Mw	3–4 V (open ckt)
Available condition	Lighted environment	Surfaces with ΔT	Hz–kHz vibration	Hz vibration	Vicinity to radiation source
Pros	High-power density well developed technology	Non-intermittent/less intermittent than alternatives	High-voltage well development technology	Well developed	Antenna can be integrated, widely available
Cons	Intermittent highly dependent on light	Low voltage needs ΔT	Highly variable output, large area, high output impedance	Bulky, low power density Low output voltage	Very sensitive to distance of the RF source

**Table 4 sensors-20-02772-t004:** Comparison of reported studies on rectennas.

Ref.	Freq. (GHz)	Gain (dBi)	Diode	Efficiencyη (%)	Dimension (cm × cm)	Load	Pin (dBm)
[44]	0.86	2.6	HSMS2850	24.2 37	11.0 × 06.0	5.0	--
[45]	0.90	1.3	HSMS2850	40.0	11.0 × 11.0	3.0	−20
[46]	2.45	6.24	SMS7630	18–32	--	3.3	−20 to −10
[47]	0.915	2	HSMS-2850	66.0	30 × 17	1.0	−06 to 32
[48]	2.45	9	HSMS-286C	10–39	75 × 40	--	15
[49]	2.45	0	SMS-7630	9–24	15.1 × 8.15	1.0	−20
[50]	0.915	−24.3	Zero-bias Schottky	--	16.0 × 14.0	2.0	−10
[51]	0.401–0.406	−23.7	--	--	34.3 × 1.95	--	8.6

**Table 5 sensors-20-02772-t005:** Comparison of previous studies of antennas.

Refs.	Freq. (GHz)	Bandwidth	Gain (dBi)	Isolation (dB)	Dimension $\left(\lambda_{0}\right)$	Efficiency (%)
[59]	30	29.13–31.10 6.57%	11.8	20	03.5 × 02.8	--
[60]	94	93.5–94.5 0.5%	26.0	15	10.3 × 10.3	57.5
[61]	60	55–69 23%	25.8	14	6.53 × 6.53	80
[62]	28	27.7–28.8 3.9%	13.5	--	6.53 × 5.93	--
[63]	60	57.2–64.2 11.5%	13.7	20	10.0 × 14.4	--
[64]	1.68– 2.75	48% |S11 | < −10 dB	8.9 ± 0.7	>37	0.78 × 0.78	82
[65]	2.2	47% |S11 | < −10 dB	~7.0	>40	1.05 × 1.05	--
[66]	1.69–2.5	31.2% |S11 | < 10 dB	~13.7	>30	1.33 × 1.33	~87.7
[67]	3.14–3.81	19.3%/20.3% (VSWR < 1.5)	~8.1	>43	0.49 × 0.49	83
[68]	1.56–2.78 1.46– 2.73	56.2%/60.6% |S11 | < 10 dB	8.05 ± 0.45	>26	0.44 × 0.44	90

Note: The operation bandwidth is the minimum bandwidth among voltage standing wave ratio VSWR < 2, AR < 3, and 3 dB gain bandwidths.

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
