# Peer review of "Synthesis, Characterization and Development of Energy Harvesting Techniques Incorporated with Antennas: A Review Study"

_sensors, 2020, doi:10.3390/s20102772_

Round 1
Reviewer 1 Report
The paper aims at making a review on the energy harvesting techniques which incorporate antennas.
Although there are many ideas and the use of tables is good, the paper lacks organization.
Additional digrams would help the presentation of ideas, for example in section 2, a diagram (as in figure 1) is needed.
Some phrases may mislead the reader. For example, in section 3.3.1 (vibration energy harvesting exploiting electromagnetic phenomenon), the authors in line 265 use the phrase "Two possible scenarios are there: linear vibration, time-varying magnetic field" (the word "there" is superfluous). It should be expressed rather as "linear vibration is exploited via time-varying magnetic field". There are not 2 different scenarios, rather a single one.
Additionally, some abbreviations are not explained (EWSN, WBAN) as well as the meaning of Table 1.
The reviewer does not recommend the publication of the work in its present form. Substantial work needs to be done to deliver better the ideas.
Author Response
Thank you for your suggestions. Attached herewith the review response.

Reviewer 2 Report
The paper is interesting as a revision contribution. The given references are comprehensive and up to date. Perhaps chapter 5 is too basic and it is mainly based on ref [80]. Sections 5.2 and 5.6 are essentially the same as the only mentioned patch antenna is a microstrip antenna.
Many spelling mistakes are found in the text. Please see the following comments. A thoroughly revision should be made.
Line 38. Needs
Line 82. RH-EH should be previously defined.
Line 118. EWSN should be previously defined.
Line 136 PETs should be removed.
Line 149 there instead of There.
Line 180 WBAN should be defined
Line 182 shown
Line 206 please include (ff) after filling factor.
Line 225 please rewrite “the power is provided by heat flow provides”
Line 235 able instead of Able.
Line 240 please remove dots in “Image. 3.”
Line 285 phi is missed. Subscript in Dem.
Line 293 vibration., please remove dot.
Lines 370-378. Please include frequency values in all the mentioned services. Moreover, these values may change depending on the ITU regions. WifI is a service that is allocated in the ISM 2.4 band and 5 GHz band.
Line 394. Table 3 needs a head explaining the meaning of different columns.
Line 418. RFEH should be defined.
Line 419. PCE should be defined.
Line 439 and 447 RF-EH should be consistent to line 418.
Line 456. miniaturization instead of Miniaturization.
Line 458 DC instead of DC
Line 461 rectena instead of Rectena. Please use always or never caps.
Line 500 please remove “the use of”
Line 503-504 Please consider removing “polarizations are the antenna polarization types”
Line 565, patch instead of path.
Line 580 and 581 please consider guided mode instead of directed mode.
Line 582 Standing Wave Ratio (SWR) instead of wave ratio.
Line 589 please include in subscripts “10”
Line 609 please remove the dot after theta. Also N is the number of loops in the case of multiple loops instead of a single one.
Line 643. Microstrip antenna and patch antenna are the same. Sections 5.2 and 5.6 should be merged.
Line 704 Qsr is not previously mentioned.
Line 727 please use V instead of v in eq. 17. Also in eq. 20.
Line 735. It instead of it. What is the meaning of DC creation DC? Maybe RF conversion to DC?
Line 762 PDC instead of PDc in eq. 21.
Line 777 thermal instead of Thermal.
Line 780 ºC instead of oC
Line 804 What dou you mean with “The selection of antennas which are using for energy harvesting based on the application like RFID, 5G, WSN, etc”?
Author Response

(The authors gave the same response as above.)

Reviewer 3 Report
Presented work presents a current topic in the field of analysis of new energy sources obtained from known and already used energy sources. The authors present a promising alternative to obtaining new energy. This energy is for energy-efficient devices from free-flowing RF signal in the atmosphere.
The authors of the paper present more possibilities for energy recovery. Most attention is paid to the relatively lesser-known radiofrequency technique. In this context, the article appeared in a larger number of procedural and technical shortcomings. This is due to a few well-known information on this topic. On the other hand, this topic can be very interesting for professional readers.
Notes for text improvement
1 The structure of the article
(a) In the Article, three chapters have the same numerical designation 5.
(In line 481) 5. Antenna
(In line 551) 5. Types of antennas
(In line 800) 5. Conclusions
(b) For review purposes, the reviewer has changed the numbering of the last chapters as follows:
(In line 481) original chapt. 5. Antenna new number. chapt. 5. Antenna
(In line 551) original chapt. 5. Types of anten. new number. chapt. 6. Types of anten.
(In line 666) original chapt. 6. RF System of ... new number. chapt. 7. RF System of ...
(In line 800) original chapt. 5. Conclusions new number. chapt. 8. Conclusions
c) It will also be necessary to change the numbering of the relevant sub-chapters. The title of the second chapter (as the only one) is written in capital letters!
2 The structure of the article (Chapter 3.3.)
In point 3.3 there are three types of conversion of vibration to electricity. These types of conversions should be named with one word. It is not appropriate to use such a simple name as a chapter name, for example. Chapter 3.3.1. "Electromagnetic". It would be preferable to use a name such as 3.3.1. "Electromagnetic conversion of vibration".
3 Table no. 3 (In-line 394)
Head of the form in the table no. 3 is not defined what is stated in each column.
4 To the text (In lines 398,399)
From the sentence “A sensor node can harvest energy from multiple revolving sources ... the question is, Is “rectanna ”a rotating source?
5 To the text (In line412)
The abbreviation PZT is in the article only shown in Figure no. 5 (In lines 412) throughout the text of this article is not simply explained.
6 To the text (In lines 418, 439 and 807)
The abbreviation RFEH appears in the body of the article in three places (In lines 418, 439 and 807), but is not explained throughout the body of the article.
7 To the text (In lines 425, 426 and 670, 671)
In the scholarly texts, it is customary to describe the individual circuits of the RF device (Rectenna) in the order shown in the block diagram. Proceed to the right from the antenna.
The text describes a “rectenna” block diagram in two places:
- Chapter No. 4 (In lines 425, 426), description of Figure No. 6,
- Chapter No. 7 (In lines 670, 671), description of Figure No. 12,
and each block description order is different.
8 To the text and Figure (In lines 425, 426 – 438, 439; 670, 671 – 680,681)
Text of Chapter No. 4 (In lines 425, 426): The rectenna is basically ...
a matching Circuit, harmonic suppression filter, antenna, voltage multiplier, load.
Chapter No. 4, Figure No. 6 (In lines 438, 439): Block diagram ...
Antenna (name is missing), Matching Circuit, Rectifier /Multiplier, Low Pass Filter, Load.
Text of Chapter No. 7 (In lines 670, 671): The circuit comprises ...
an antenna, a voltage regulator, an impedance matching network, a multi-stage rectifier circuit, a load capacitor.
Chapter No. 7, Figure No. 12 (In lines 680, 681): General Block Diagram ...
Transducer, Impedance matching, Multi-Stage RF-to-DC Rectifier, DC-to-DC Regulator, Load.
9 Figure 12 (In line 680)
Figure 12 is a block diagram of energy conversion, consisting of six elements, but no chapter 7 the accompanying text, (In lines 669 ÷ 671) is given only five elements.
10 To the text (In line 693)
In the sentence (Inline 693), three parameters at the same level are labelled differently. The (Crec) parameter has double-sided parentheses, the Rrec) parameter has a one-sided parenthesis, and the Zrec parameter has no parenthesis.
11 To the text (In lines 693, 697)
In the sentence (Inline 693), the parameters (Crec, Rrec, Zrec) are labelled "rec" as the subscript. The similar designation is also in formula no. 14 (Inline 695). Subsequently in the sentence (In line 697) and in figures no. 13 and 14, the same parameters have the designation “rec” not as subscript (C rec, R rec) but as equivalent fonts.
12 Formula 14 (In line 695)
Formula No. (14) is numbered as a single formula throughout the article cursive form of text.
13 Formula 16 (Inline 704)
The parameter Qsr in formula (16) is not explained in the text.
14 To the text (In lines 705)
Sentence (In lines 705) "Where VS and VIN are the input voltage amplification and the rectifier input voltage" is to respect 16 and Figure 16 vaguely defined.
15 To the text (In lines 717 ÷ 720)
Chapter 7.2 explains the diode principle in the text (In lines 717 ÷ 720). In the first part of the sentence, it is correctly stated that the diode has a non-linear correlation between current and voltage. The second part of the sentence explains this. Why in explaining the correlation between I and U in the second part of the sentence does not current already not stated?
16 Formula 17 and 18 (In lines 717 ÷ 720)
a) In formula no. 17 and no. 18 defines one and the same value of the current I, which are given differently in formula no. 17 - I(V) and in formula no. 18- I(V).
b) In addition, in the exponent in formula no. (17) is the expression (αv) → (eαv) and in formula no. (18) is the expression (Θv) → (eΘv). Which of these two terms is correct?
c) In formula to no. (17) there is a small "v" (eαv) in the exponent, but a large "V" is explained in the text.
d) In formula to no. (18) is not explained in the text which is "Θ".
17 Formula 21 (In lines 717 ÷ 720)
In formula no. 21, the parameters (PDc, out, PRF,in) are written in the same way. They are marked differently. In the following explanatory text they are written in the same way (PDC,out, PRF,in), but in a different way than in formula 21.
18 Figure 18(b) (In line 750)
Why not in Figure 18(b) input voltage Vin harmonic signal displays symmetrical zero value?
Some remarks to the conclusion
1 From the presented results of Table no. 3, it can be stated that the range and level of the captured signal increases as the carrier frequency decreases.
2 An equivalent rectenna circuit diagram is shown in four figures (13, 14, 15, 16). It is well known that an important role of a rectifier is played by a diode (Chapter 7.2). Why isn't the diode shown in either of these images? The R, C circuit itself will not direct or increase the voltage!
3 In conclusion, the authors could state: What is the legislation about such consumption of energy in the vicinity of the transmitting antenna?
Author Response

(The authors gave the same response as above.)

Round 2
Reviewer 1 Report
I would like to thank the authors for taken in consideration the suggestions made by the reviewers.
Author Response
Thank you for your valuable suggestions.

Reviewer 2 Report
Thank for your reply and comments.
I have found the followin spelling mistakes:
Line 185: "sown in Figure.1. In addition, Table 4. Shown the basic characteristics of the method of scavenging" should be "shown in Figure.1. In addition, Table 4. shows the basic characteristics of the method of scavenging"
Line 239: "other energy-harvesting techniques are considered [22]. Image 3. Displays the thermoelectric" should be "other energy-harvesting techniques are considered [22]. Image 3. displays the thermoelectric"
Line 358: "Figure 5 shown the schematic representation" should be "Figure 5 358 shows the schematic representation"
Line 414: According to figure 6, TEG should be TE.
Line 439: Figure 7. "Rectifire" should be Rectifier.
Line 562: Figure 9. "Paracitic" should be Parasitic.
Line 615: TE"10" should be TE10
Line 812. Please consider to change your original sentence "The selection of antennas which are using 812 for energy harvesting based on the application like RFID, 5G, WSN, etc" for the explanation in your reply "Antennas are one of the essential parts of the developed energy harvesting system whereas the performance of those antennas such as efficiency, gain and radiation patterns is critical to improve the overall performance. Due to this reason, it very important to select the suitable antenna based on the application" and the "For instance" instead of "For instant"
Author Response
Thank you for your suggestions.
